

# Assimilating bio-optical glider data during a phytoplankton bloom in the southern Ross Sea

Daniel E. Kaufman[1], Marjorie A. M. Friedrichs[1], John C. P. Hemmings[2,3], Walker O. Smith Jr.[1]

[1]Virginia Institute of Marine Science, College of William & Mary, Gloucester Point, VA, USA

[2]Wessex Environmental Associates, Salisbury, UK
[3]now at Met Office, Exeter, UK

*Correspondence to:* Daniel E. Kaufman (dekaufman@vims.edu) and Marjorie A. M. Friedrichs (marjy@vims.edu)

**Abstract.** The Ross Sea is a region characterized by high primary productivity in comparison to

other Antarctic coastal regions, and its productivity is marked by considerable variability both

spatially (1-50 km) and temporally (days to weeks). This variability presents a challenge for

inferring phytoplankton dynamics from observations that are limited in time or space, which is

often the case due to logistical limitations of sampling. To better understand the spatiotemporal

variability of Ross Sea phytoplankton dynamics and determine how restricted sampling may

skew dynamical interpretations, high-resolution bio-optical glider measurements were

assimilated into a one-dimensional biogeochemical model adapted for the Ross Sea. Assimilation

of glider data using the micro-genetic and local search algorithms in the Marine Model

Optimization Testbed improves model-data fit by ~50%, generating rates of integrated primary

production of 104 g C $m^2$ $y^{-1}$ and export at 200 m of 27 g C $m^{-2}$ $y^{-1}$. Assimilating glider data from

three different latitudinal bands and three different longitudinal bands results in minimal changes

to the simulations, improves model-data fit with respect to unassimilated data by ~35%, and

confirms that analyzing these glider observations as a time series via a one-dimensional model is

reasonable on these scales. Whereas assimilating the full glider data set produces well-

constrained simulations, assimilating subsampled glider data at a frequency consistent with

cruise-based sampling, results in a wide range of primary production and export estimates. These

estimates depend strongly on timing of the assimilated observations, due to the presence of high

mesoscale variability in this region. Assimilating surface glider data subsampled at a frequency

consistent with available satellite-derived data results in 40% lower carbon export, primarily

resulting from optimized rates generating more slowly sinking diatoms. This analysis highlights

the need for strategic consideration of impacts of data frequency, duration, and coverage when



combining observations with biogeochemical modeling in regions with strong mesoscale
variability.



## 1 Introduction

35        Phytoplankton blooms in the Ross Sea are responsible for some of the highest rates of productivity in the Southern Ocean (Arrigo et al., 2008), and yet the phytoplankton assemblage exhibits considerable spatiotemporal variability (DiTullio and Smith, 1996; Hales and Takahasi, 2004; Smith et al., 2010). This heterogeneity, and the spatial/temporal limitations of observations due to logistical challenges of sampling, may affect the inferred phytoplankton dynamics and

produce biases in productivity or export estimates. The magnitude of the underlying ecosystem variability that contributes to these potential biases is not well understood, nor is it well known how the use of different observational platforms in the Ross Sea might affect the inferred dynamics. Acquiring data with an appropriate resolution is important for assessing phytoplankton variability in the Ross Sea (Hales and Takahashi, 2004).

45        Over the past several decades, biogeochemistry in the Ross Sea has been observed by ship and satellite, providing data at different temporal and spatial resolutions. Since Ross Sea phytoplankton became a focus of scientific research in the late 1970s, water column measurements have primarily come from research vessels (e.g., El-Sayed et al., 1978; Smith and Nelson, 1985; Vaillancourt et al., 2003). Typically, sampling stations are separated by tens of

kilometers (Hales and Takahashi, 2004), and although vessels may return to resample a station, they typically do not return more than once or twice in a single year. During the 1990s, the use of remote sensing was expanded to look more closely at the Ross Sea bloom (Arrigo and McClain, 1994), and satellite retrievals have continued to provide valuable insights into characteristics of the phytoplankton assemblage (Arrigo et al., 1998; Arrigo and van Dijken, 2004; Peloquin and

Smith, 2007; Schine et al., 2015). Satellite observations offer a synoptic view of spatial regions at frequencies that are within the time scale of biological changes (e.g. growth); however, the presence of sea ice and clouds often obscures remote-sensing measurements in the Ross Sea (Arrigo et al., 1998).

        At the mesoscale (days-weeks, 1-10 km), gliders are a relatively new and effective means

to characterize phytoplankton variability, and the development of ice-avoidance algorithms has enabled the use of gliders in the Ross Sea for these purposes. For example, a glider equipped with bio-optical sensors was directed along a section near 76° 40′ S in austral summer 2010 - 2011 and provided valuable estimates of biomass variability on short time scales (Kaufman et al., 2014). Estimates of the POC:Chl ratio from the glider optical sensors suggested a transition from



a *Phaeocystis antarctica* to a diatom-dominated assemblage over several days (Kaufman et al.,

2014; Thomalla et al., 2017). Moreover, Jones and Smith (2017) used glider observations from

austral summer 2012-2013 to distinguish three phases of the Ross Sea bloom and identified high

frequency (hours) associations between wind-driven mixing and biomass. A perennial challenge

when using glider data (as well as ship-based data), however, is separating the effects of time and

space (Kaufman et al., 2014; Little, 2016).

Numerical models are another approach for examining phytoplankton variability in the

remote Ross Sea, providing an effective means for coordinating knowledge and understanding

the underlying system complexities (Leonelli, 2009; Vallverdú, 2014). Furthermore, numerical

simulations offer the ability for experimental manipulations that would be impractical or

impossible in the real system. Such manipulations were implemented in the scenario experiments

described by Kaufman et al. (2017a) to investigate how projected climate changes might alter the

dynamics of the phytoplankton assemblage. These experiments showed that earlier availability of

low light resulting from sea ice reduction was the primary driver of projected increases in

production and export and composition change over the next century.

Data assimilation, which refers to methodologies that systematically combine a

mathematical model with observations, is often used in biogeochemical applications (Hofmann

and Friedrichs, 2001, 2002) to improve estimates of model parameters that are frequently poorly

known (Lawson et al., 1995, 1996; Matear, 1995; Fennel et al., 2001; Friedrichs, 2002; Schartau

and Oschlies, 2003; Hemmings et al., 2004; Doron et al., 2013; Xiao and Friedrichs, 2014a,b;

Melbourne-Thomas et al., 2015; Gharamti et al., 2017, Schartau et al., 2017). This entails a

smoothing or optimization procedure, in which elements of the model are adjusted to minimize

differences between the model output and the observations. Typically, an aggregate measure of

the differences between observations and model output is provided by calculation of a cost

function, defined as the model-data misfit, and an optimization algorithm searches for model

parameters that minimize the value of this cost function.

In this study, data assimilation is used to obtain an optimal representation of Ross Sea

lower trophic levels. Specifically, observations from an autonomous glider are assimilated into a

biogeochemical model of the Ross Sea (Kaufman et al., 2017a) to better understand the spatial

and temporal variability of phytoplankton in this region. Assimilation experiments also examine



how the space and time characteristics of observational sampling frequency impacts the ability of
observations to produce optimal system representations.

## 2 Methods

### 2.1 One-dimensional biogeochemical model

Numerical experiments were conducted with the Model of Ecosystem Dynamics, nutrient

Utilisation, Sequestration and Acidification for the Ross Sea (MEDUSA-RS; Kaufman et al.,
2017a), a regionally adapted version of MEDUSA-1.0 (Yool et al., 2011). Three phytoplankton
groups are represented in the MEDUSA-RS model: colonial *P. antarctica*, solitary *P. antarctica,*
and diatoms. Phytoplankton growth in the model is temperature dependent as well as limited by
light and nutrient availability. Colonial *P. antarctica*, diatoms, and detritus all sink at distinct

rates. The model handles the sinking of large detrital particles implicitly as a fast-sinking group
to avoid issues related to the scale of the model time step and to avoid the need for an additional
tracer. A ballast scheme is used to allow inorganic materials to "protect" a variable fraction of
the sinking organic material from degradation. A full description of the model and its set-up
within the Marine Model Optimization Testbed (MarMOT; Hemmings and Challenor, 2012), as

well as the physical forcings derived from glider observations, are documented in Kaufman et al.
(2017a,b).

### 2.2 Data for assimilation

In situ observations used for the assimilation experiments came from an iRobot Seaglider
equipped with a Wet Labs ECO Puck sensor and are available in the BCO-DMO data repository

(http://www.bco-dmo.org/dataset/568868). Glider dives from 22 November 2012 to 01 February
2013 covered a horizontal area spanning 76.83 - 77.44 ˚S and 168.9 - 171.97 ˚E (Fig. 1). Data
spanning the upper 200 m of the water column were binned into hourly, 5-m vertical bins.
Concentrations of chlorophyll (Chl) and particulate organic carbon (POC) were derived,
respectively, from fluorescence and optical backscatter counts measured by the sensor and

converted using regression equations (Kaufman et al., 2017a). These bio-optical quantities were
used for calculating model-data misfits during assimilation.

### 2.3 Cost function

The 'cost function' ($J$), defined as a measure of misfit between a particular model
simulation and observational data, is computed as a weighted average of the squared differences

between simulated and observed values:




$$\mathcal{J} = \frac{1}{N}\sum_{i=1}^{N}\left(\frac{1}{\sigma_{chl}^2}\left(x_{i,chl} - y_{i,chl}\right)^2 + \frac{1}{\sigma_{poc}^2}\left(x_{i,poc} - y_{i,poc}\right)^2\right)$$

where $N$ is the number of observation points, $\sigma$ is the standard deviation of each variable, $x_i$ is the simulated value of either chlorophyll or POC at the $i$th observation point and $y_i$ is its observed value. Model parameters were optimized in MarMOT by finding the minimum of this cost function through a combination of the micro-genetic algorithm and Powell's non-gradient

direction set algorithm, as described below.

**2.4 Implementation of micro-genetic algorithm and direction set algorithm**

Genetic algorithms, including the micro-genetic algorithm [µGA], are a subtype of computational methods known as evolutionary algorithms, so-called because of their inspiration from, and metaphoric relationship to, biological evolution. Described using this metaphor, a

genetic algorithm procedure modifies a population of candidate solutions over successive generations by variation and selection processes to converge on a single solution or solution area. GAs have several advantages for optimization, including their intrinsic parallelism, suitability for systems with multiple local minima, and their generalizability (Bajpai and Kumar, 2010; Ward et al., 2010). The µGA uses three steps to transition from one generation to the next, described

following the biological metaphor as: selection, crossover, and resampling (Krishnakumar, 1990; Črepinšek et al., 2013). An advantage of the µGA is its reduced risk of premature convergence, resulting from reinitializating after each convergence, and generating new random populations while maintaining the best fit individual from the previous set (Schmitt, 2001).

In this µGA implementation, optimizations begin with a population of five individual

parameter sets randomly generated for the first µGA generation. An evaluation of the cost function for each model solution indicates the 'fitness' of each individual. A binary tournament procedure is then followed to select parents from this population for the next generation. The most-fit individuals (i.e., those with the lowest cost function values) are paired with one another and undergo recombination of the bits representing parameter values. After each generation, the

proportion of bits differing from those of the fittest individual is calculated to determine whether the population can be deemed converged (though this does not necessarily indicate closeness in parameter space). After the threshold for convergence has been achieved, the population is reinitialized to random individuals, although the fittest individual is maintained. The µGA is



terminated upon the first convergence occurring after a minimum number of generations has
been reached.

Once convergence has been achieved after a minimum number of μGA generations,
Powell's non-gradient direction set algorithm performs a local search using the μGA solutions as
starting points. The direction set method performs sequential minimizations in iterative
directions, updating the search direction after each iteration (Powell, 1964; Press et al., 1992).
Although the μGA is well suited for global search problems partly because of its stochasticity,
Powell's direction set algorithm is well suited to searching for a local optimum. Brent's method,
which combines root-bracketing with secant and inverse quadratic interpolation (Brent, 1973), is
used to numerically locate cost minima between neighboring function evaluations along each
direction identified by the Powell algorithm. The direction set algorithm stops when a cost
function minimum is located or when a maximum number of iterations is reached. The optimized
parameter values are those that generated the cost function minimum.

**2.5 Selection of parameters to be optimized**

Even though it is tempting to try to optimize all parameters in a model, uncertainty in the
parameter estimates increases as the number of parameters optimized increases (Friedrichs et al.,
2007; Ward et al., 2010). Although the optimization of more parameters generally lowers the
assimilated cost, the increasing potential for equifinality with more parameters means the
optimization may find equivalent low-cost solutions with substantially different parameter
values. Therefore, before assimilating observations and optimizing parameters, a subset of "free"
or "optimizable" model parameters must be chosen. In this study, the parameters to be optimized
are selected through a three-step process: defining a range of permitted values for every
parameter (Sect. 2.5.1), identifying the parameters to which model outputs are most sensitive
(Sect. 2.5.2), and evaluating how many of these sensitive parameters can be reasonably
optimized when assimilating the available data (Sect. 2.5.3). Initial values for each parameter,
prior to the assimilation, were set to values identified in Kaufman et al. (2017a).

**2.5.1 Parameter ranges**

Upper and lower bounds of the allowable range for each free parameter were defined
loosely following Hemmings et al. (2015). Bounds were set to be geometrically symmetric
(factor of four for rates; factor of five for half-saturation concentrations) around the initial
values. For fractional parameter values, limits were set to +/- 0.25 their initial values, although



not allowed to exceed 0.05 or 0.95. Ranges for parameters not expressed as fractions were log-

transformed for sampling purposes.

**2.5.2 Sensitivity Analysis**

        Parameters to which model outputs are highly sensitive are important and useful to

optimize. In contrast, it is futile to optimize parameters to which model outputs of interest are not

sensitive; no amount of varying these parameters will result in improved model performance.

Therefore, the first criterion used to designate a parameter as optimizable was the sensitivity of

model outputs to the values of that parameter. Model sensitivities were evaluated for assimilated

variables (Chl and POC) and carbon fluxes of interest (primary production (PP) and carbon

export at 200 m). To quantify the sensitivities of these outputs to each of the 80 parameters in the

model, a series of runs were conducted following the approach of Hemmings et al. (2015). Each

run used a unique sample of parameter values drawn from within the specified parameter ranges

(Section 2.5.1) using a Latin hypercube. This approach provides more even coverage of the

parameter space than Monte Carlo sampling methods that can result in clustered values and

unsampled regions. One thousand values were drawn from sequential intervals throughout the

range for each parameter. Using this technique, unique parameter sets were constructed such that

over the course of all runs, the full range of values for each parameter was represented.

        The model was run 1000 times, each time using one of the unique parameter sets

resulting from Latin hypercube sampling of the full parameter space. Sensitivity was quantified

by evaluating the amount of variance in the output diagnostics explained by each parameter (i.e.,

by computing the coefficient of determination ($r^2$) between each parameter and each of the four

output variables of interest; Fig. 2). All four model outputs (Chl, POC, PP, and export) were

most sensitive ($r^2 \geq 0.01$) to attenuation of blue-green light by phytoplankton pigments, diatom

maximum growth rate, and C:Chl ratio for solitary *P. antarctica*. Three additional parameters

(maximum growth rate of *P. antarctica* colonies, maximum growth rate of solitary *P. antarctica*,

and microzooplankton maximum grazing rate) exhibited $r^2 \geq 0.01$ for both chlorophyll and POC.

The 21 parameters with $r^2 \geq 0.01$ (Fig. 2) were selected for further evaluation (Sect. 2.5.3).

**2.5.3 Using twin experiments to select optimizable subset**

        After selecting the 21 potentially optimizable parameters, Numerical Twin Experiments

(NTEs) were conducted to evaluate the extent to which known values of sensitive parameters

could be recovered given the data available for assimilation. The implementation of NTEs



involves four primary steps (Hofmann and Friedrichs, 2001). First, the chosen model is run forward in time to create a simulation using a known, "true" parameter set. Second, output from this simulation is sub-sampled to create a so-called "synthetic" data set. Third, the synthetic dataset is then assimilated to optimize model parameters. Fourth, the optimized parameter set is

compared to the true parameter set. The assimilation is successful if the optimized values recover the true parameters used to generate the assimilated synthetic data.

The procedure followed here for determining the subset of optimizable parameters is similar to that used by Friedrichs et al. (2007). First, a reference simulation was generated using the initial parameter set, and chlorophyll and POC estimates from this reference simulation were

subsampled to generate a synthetic data set. Starting with a parameter space defined by the set of 21 parameters deemed sensitive in the Latin hypercube tests (Fig. 2), a series of sequential NTEs was then performed with a progressively smaller number of optimized parameters: after each NTE, the optimized parameter that was most different from its 'true' value was removed from the optimizable parameter set. Thus, after each NTE the number of optimized parameters was

reduced by one. The series of NTEs was evaluated to identify the largest parameter set for which the original parameter values were recoverable and the cost function remained essentially zero. From this analysis (Fig. 3), it was determined that optimizing eight parameters would be ideal (Table 1), because values of these eight parameters were recovered much better than larger parameter sets and model-data misfit (cost) remained low.

**2.6 Assimilation experiments**

The μGA optimization procedure was used to assimilate glider data in two sets of experiments that explored aspects of spatiotemporal variability and data availability. Estimates of depth- and time-integrated PP and time-integrated carbon export at 200 m were computed from the full model simulation in each experiment.

**2.6.1 Experiment #1**

The first set of experiments examined the differences in model simulations resulting from assimilating Chl and POC data from different spatial regions. In Experiment #1a, glider observations were assimilated from the upper 50 m of the full temporal and spatial domain, referred to hereafter as the "Full Assimilation" case (Table 2). (Comparisons showed only minor

differences between assimilating data from the upper 50 m vs. the full upper 200 m). Observations from different spatial areas of the glider track were also assimilated. Observations



from the glider track were divided into three latitudinal bands (Northern, Central, and Southern bands) as well as into three longitudinal areas constituting Eastern, Central, and Western bands. Glider data from each of these three latitudinal and longitudinal bands were assimilated in

Experiment #1b and #1c, respectively (Table 2, Fig. 4), resulting in three cost functions for each of these esperiments.

**2.6.2 Experiment #2**

The second set of experiments investigated the assimilation of data at different resolutions mimicking different data sources. In Experiment #2a, glider data were subsampled at

~12-hour intervals (Table 2). The subsampling was repeated 12 times, with each iteration offset from the previous by +1, 2, 3... 11 hours, to generate a series of 12 glider observation sets. The assimilation of these 12 time series yields the "Glider Assimilation" case. In Experiment #2b, glider data were subsampled at a reduced temporal resolution similar to cruise sampling (Table 2). Sampling during cruise missions often takes place for a few days in one location before

moving elsewhere, and the ship sometimes returns to the first location after a number of weeks. To roughly mimic this sampling pattern, daily vertical profiles (again down to 50 m) were assimilated for three days in a row, starting from the first day of available glider data (22 Nov), and then three days of data were assimilated two weeks later. Shifting this pattern forward one week at a time generated a series of eight cruise-based observation sets for assimilation in this

"Cruise Based Assimilation" case. In Experiment #2c, glider data were assimilated only from the upper 5 m surface layer to produce a data set resembling satellite-derived data. These data were then subsampled at two-week intervals, to represent typical data return from remote sensing observations of ocean color in the Ross Sea, where the availability of satellite image retrieval is frequently limited by excessive, though variable, cloud cover (Arrigo and van Dijken, 2004). The

two-week subsampling pattern covered the entire period of glider data (22 Nov - 1 Feb), and was sequentially shifted forward one day at a time to generate a series of 14 satellite-based observation sets for assimilation in this "Satellite-Based Assimilation" case (Table 2).

**2.7 Predictive Cost Assessment**

In addition to the assimilative cost ($\mathcal{J}_A$) calculated during the optimization procedure

using assimilated data, a predictive cost ($\mathcal{J}_P$) was calculated to assess model-data misfit computed using the unassimilated data in each experiment. Because predictive costs represent model-data misfit from unassimilated data only (Friedrichs et al., 2006; Ward et al., 2010), it is





an objective measure of the skill of an optimized model in reproducing observations at different
points in time or space (Gregg et al., 2009). In this case, the aim of these experiments is to assess

the skill of each optimized simulation regardless of which subset of the available data is
assimilated. By computing the mean and median predictive cost for each experiment (other than
the Full Assimilation case), the skill of the resulting simulations can be compared directly with
one another.

## 3 Results

**3.1 Experiment #1**

Assimilation of the glider data over the full temporal and spatial domain (Full
Assimilation case) improves the model-data fit of both Chl and POC (Fig. 5a,b) and reduces the
cost by nearly half (47%) compared to the a priori simulation without assimilation (Table 3).
Average Chl and POC concentrations in the upper 50 m are both slightly lower (8% and 12%,

respectively) in the optimized simulation. The contribution of each phytoplankton group to total
chlorophyll remains similar to the No Assimilation case (Fig. 6a,c), but colonial *P. antarctica*
carbon is lower and diatom carbon is higher in December and early January (Fig. 6b,d).
Compared to the No Assimilation case, PP is only slightly lower (7%), whereas export flux is
nearly 50% higher (Table 3; Fig. 7). Compared to their initial values, colonial *P. antarctica*

parameters change the most as a result of the optimization, with reductions between 40-70% for
the colonial *P. antarctica* maximum growth rate, maximum sinking rate and C:Chl ratio (Table
4). In contrast, the diatom maximum growth rate and C:Chl ratio increased (~10% and 20%
respectively).

Chlorophyll and POC time-series exhibit only minor differences between latitudinal band

experiments when data from the northern, central, and southern sections are assimilated
independently (Fig. 5c,d) or when data from the eastern, central western sections are assimilated
(Fig. 5e,f). Specifically, the optimal simulations for Chl and POC exhibit similar seasonal cycles
across the three latitudinal and longitudinal bands, with only slightly higher Chl and POC
concentrations when assimilating data from the southern band (Fig. 5c,d) and higher Chl from

the western band (Fig. 5e,f). Mean costs are much lower for the latitudinal and longitudinal
experiments than for the No Assimilation case, and only slightly higher than the Full
Assimilation case (Table 3). This indicates that data sampled from within only one spatial band
improved the match between modeled and observed variables in the unassimilated areas as well.



Average estimates of PP and export in both the latitudinal and longitudinal experiments are only

slightly less (< 5%) than the Full Assimilation estimate (Fig. 7, Table 3).

**3.2 Experiment #2**

Assimilation of data subsampled at a frequency one-twelfth that of the original glider data (Expt. 2a) results in twelve model simulations, all of which are similar to the Full Assimilation case, with Chl and POC time series closely following the observed seasonal pattern (Fig. 8a,b).

Mean assimilative and predictive costs in the Glider Assimilation case are close to the cost of the Full Assimilation case (Table 3). Mean PP and export estimates are also close to estimates from the Full Assimilation case. The mean optimal parameter values obtained from the Glider Assimilation case are generally within one standard deviation of the optimal values from the Full Assimilation case (Table 4).

Assimilation of data subsampled with a frequency typical of cruise observations (Expt. 2b) results in a wide range of solutions, with several Chl and POC time series exhibiting markedly different peak bloom timings (Fig. 8c,d). Two of the solutions yield substantially higher concentrations of POC in November, and Chl peaks range from mid-November to early January. The mean predictive cost from this experiment (1.24) is roughly three times the

assimilative cost for the Full Assimilation case (0.41) and three times the predictive cost for the Glider Assimilation case (0.43; Table 3). The PP estimates from the Cruise-based Assimilation case span a broad range (92 to 156 g C m$^{-2}$ y$^{-1}$) around the Full Assimilation estimate but are generally higher (Fig. 7a). This experiment similarly yields a very large range of export estimates (11 to 33 g C m$^{-2}$ y$^{-1}$) encompassing the results from Experiment #1 (Fig. 7b). Optimal parameter

values obtained from the Cruise-based Assimilation case are generally less well constrained (higher standard deviations) than the Glider Assimilation case (Table 4).

Assimilation of data subsampled as satellite-based observations from the surface layer (Expt. 2c) results in Chl and POC concentrations generally higher than the Full Assimilation case (Fig. 8e,f). The predictive costs are similar on average to those of the Cruise-based Assimilation

experiment; however, there is less variation (Table 3). The median integrated production is higher (9%) than the Full Assimilation estimate and the Cruised-based Assimilation estimate (Fig. 7a; Table 3); however, the range of PP estimates for this Satellite-based Assimilation case is smaller than those for the Cruise-based Assimilation case (Fig. 7a). Most notably, despite generally higher PP and higher POC concentrations, carbon export from the Satellite-based



Assimilation case is substantially lower (41%) than the Full Assimilation estimate (Fig. 7b;
Table 3). In fact, export estimates from individual runs in this experiment are all lower (-19% to
-56%) than the Full Assimilation estimate (Fig. 7b). Again, the range of export estimates is
smaller for the Satellite-based Assimilation than for the Cruise-based Assimilation. When
assimilating data at a resolution similar to that of satellite-based observations, mean optimal

parameter values were similar to those obtained in the Glider Assimilation and Cruise-based
Assimilation cases, with the exception of the fast detritus sinking fraction for diatoms, which was
significantly lower in the Satellite-based Assimilation case ($0.62 \pm 0.14$) than in the other
experiments (Glider Based Assimilation Case: $0.86 \pm 0.05$). In contrast to this sinking parameter
for mortality from diatoms, the mean maxmimum sinking rate of colonial *P. antarctica* in the

Satellite-based case was not significantly different than its value in either the Full Assimilation
or Cruise-based cases (Table 4). Standard deviations of optimal parameters for the Satellite-
based Assimilation case were generally similar to or lower than those for the Cruise-based
Assimilation case, except for the C:Chl ratio for diatoms, which produced a very high optimal
value and was particularly poorly constrained ($375 \pm 187$ gC gChl$^{-1}$; Table 4).

**4 Discussion**

**4.1 Ross Sea simulation resulting from assimilation of glider data**

Data assimilation is a valuable tool for efficiently utilizing limited observational data in
remote regions like the Ross Sea. In this study, glider data consisting of both fluorescence-
derived chlorophyll and backscatter-derived POC were assimilated into a one-dimensional

marine biogeochemical model developed for the Ross Sea. Eight ecosystem parameters,
including phytoplankton rates and C:Chl ratios, were optimized resulting in a simulation with a
50% reduced model-data misfit. This optimal simulation yielded lower *P. antarctica* carbon
concentrations and higher diatom carbon concentrations, resulting in higher carbon export
compared to those generated by the initial hand-tuned simulation (Kaufman et al., 2017a),

despite slightly lower estimates of overall annual primary production. Changes in chlorophyll
concentrations of diatoms and *P. antarctica* were minor. This optimal simulation was obtained
largely via changes in the C:Chl ratios: the colonial *P. antarctica* ratio of C:Chl was lower and
the diatom C:Chl was higher than in the original simulation. Although modified from their initial
values, the relative differences between these optimized C:Chl ratios for *P. antarctica* and

diatoms are consistent with shipboard measurements of C:Chl ratios, which found higher C:Chl



in diatom-dominated waters compared to *P. antarctica*-dominated waters: ~200 vs. 90 g C g Chl⁻¹ (DiTullio and Smith, 1996), and ~50-100 vs. 20-50 g C g Chl⁻¹ (Mathot et al., 2000). Although the authors are not aware of any specific estimates in the literature for the fraction of diatom mortality that becomes fast-sinking detritus, other optimal rate paramaters are consistent with

those previously reported in the literature. For example, the optimized growth rates (0.29 - 0.4 m d⁻¹) are similar to measured values in the Ross Sea (Smith and Gordon, 1997; Smith et al., 1999; Mosby and Smith, 2015), and the optimized sinking rate of *P. antarctica* colonies (14 m d⁻¹) is similar to previous estimates (Asper and Smith, 1999; Asper and Smith, 2003; Smith et al., 2011).

**4.2 Spatial variation within the glider track**

Phytoplankton in the Ross Sea exhibit both spatial and temporal variability. Cruise transects across the continental shelf show a marked spatial variability in both the east-west and north-south direction over short periods of time (Smith et al., 2013). Within the Ross Sea Polynya, ship-based observations show biochemical gradients that suggest patchiness of

phytoplankton dynamics on the mesoscale (Hales and Takahashi, 2004; Smith et al., 2017). Nutrient pools have been found to exhibit gradients from both north to south and east to west (DiTullio and Smith, 1996; Sedwick et al., 2011; Smith et al., 2013; Marsay et al., 2014), and phytoplankton assemblage composition is not necessarily uniform across longitudes (DiTullio and Smith, 1996; Garrison et al., 2003; Smith et al., 2013). In addition, cold and fresh eddies

have been observed along the ice shelf edge potentially reshaping the phytoplankton assemblage on short time (<10 days) and space (<20 km) scales (Li et al., 2017).

When analyzing glider data in regions characterized by high mesoscale variability, it is often not apparent whether observed patterns represent spatial or temporal variability. As Rudnick (2016) discusses, "Because gliders can occupy lines, their data can be viewed as

traditional sections, such as those measured from a ship. However, because high-frequency variability is projected onto a spatial structure, it is sometimes more convenient to think of the data as a time series from a mooring." This ambiguity led Kaufman et al. (2014) to concede "both spatial and temporal gradients may have played a role in the observed variability" when analyzing physical-biological relationships from glider data in the southern Ross Sea.

Although both temporal and spatial gradients may be present, observations can be presented as either primarily spatial or temporal patterns with simple tests guiding the decision.



For example, a comparison of means and standard deviations across spatial sections and time periods was previously used to identify time as the dominant dimension of variability in the 2012-2013 glider observations (Jones and Smith, 2017). In this study, a similar conclusion was

reached, using a very different methodology. The assimilation of glider data from nine different sub-areas of the study region (separated latitudinally or longitudinally by ~20 km) indicated that the seasonal cycle is similar in phase throughout the region of the glider track. The assimilation of glider data from each of the nine regions yielded similar estimates of POC and Chl, generally within the variance of the glider observations (gray areas of Fig. 5c-f), and similar estimates of

temporally averaged primary productivity and export. This further supports the approach of using the glider data as a time series and suggests that temporal patterns at this scale play a greater role than spatial patterns in structuring variability of the phytoplankton assemblage. Moreover, the similarity between predictive and assimilative costs when assimilating the latitudinal and longitudinal bands of data suggests that the parameters are not being over-fit for

these experiments. Thus, temporally resolved observations in any of these regions might be expected to provide similar constraints on modeled temporal patterns of the phytoplankton.

**4.3 Differences between assimilating glider, satellite-derived, and cruise-based data**

Results from experiments that assimilated data at different spatial and temporal resolutions suggest that assimilating only surface observations, as are typically available from

remote-sensing platforms, underestimates carbon export and more weakly constrains estimates of productivity relative to assimilation of depth-resolved glider data. The lower estimates of carbon export occurred because the optimal diatom fraction for fast-sinking detritus obtained via the assimilation of surface-only data ($0.62 \pm 0.14$) was significantly lower than that obtained via the assimilation of data throughout the upper 50 m (Expt. 2a: $0.86 \pm 0.05$; Expt. 2b: $0.86 \pm 0.11$).

Experimental results also indicate that the assimilation of satellite-derived data provides a weaker constraint on productivity estimates, as seen by the larger range of estimates ($114 \pm 11$ gC m$^{-2}$ y$^{-1}$), as compared to the assimilation of glider data ($104 \pm 2$ gC m$^{-2}$ y$^{-1}$). Although not statistically significant, the higher productivity estimates generated by the assimilation of satellite-derived data is consistent with those of Gregg (2008), who found that assimilation of

satellite-based chlorophyll estimates into a three-dimensional global biogeochemical model overestimated primary production. In contrast, results from assimilating satellite-derived



chlorophyll concentrations into a one-dimensional model in the equatorial Pacific produced underestimates of primary productivity compared to in situ observations (Friedrichs, 2002).

Although both chlorophyll and POC were assimilated in the present study, chlorophyll
alone has been the dominant satellite data product used in biogeochemical assimilation, although other data types are available and can impact the assimilation results. For instance, a study investigating the assimilation of different types of satellite-derived data, including POC and size-fractionated chlorophyll, found that assimilation of satellite-derived POC estimates worsened the model estimates of chlorophyll, whereas the assimilation of chlorophyll did not substantially
impact the POC estimates (Xiao and Friedrichs, 2014b). Additionally, satellite-based sampling bias could be reduced by concurrently assimilating export flux data derived from sediment trap measurements (Friedrichs et al., 2007), or by assimilating satellite measurements such as remote-sensing reflectance directly (Jones et al., 2016). It is also worth noting that when assimilating actual satellite data, the biases suggested by this study resulting from assimilation of only surface
data would be compounded with biases inherent in the satellite retrieval algorithms (Saba et al., 2011; Stukel et al., 2015).

Assimilating cruise-based data in the highly variable Ross Sea may also yield potentially large errors in primary production, as well as in carbon export estimates, depending on which specific days are sampled. Estimates of bloom timing from the assimilation of cruise-based
observations may also vary substantially (Fig. 8c,d). This echoes the results of a series of reduced resolution data interpolations, from which Hales and Takahashi (2004) reported that cruise-based observations in the Ross Sea were likely able to capture average conditions well, but miss some mesoscale phenomena. Likewise, a subsampling analysis of physical-biological correlations from 2010 Ross Sea glider data demonstrated the possibility of lower resolution data
obscuring or biasing biogeochemical interpretations (Kaufman et al., 2014). The results provided by the data assimilative study described here can be used to help guide decisions of when and how long to sample certain locations in the Ross Sea; this is especially important given the limitations of ship-based sampling in such a remote region (Smith et al., 2014). In fact, the use of data collection from other sampling platforms may decrease the pressure to conduct repeated
transects by ship, and allow limited vessel-time to be used for more thorough process-based investigations uniquely-suited for research vessels.





## 5 Summary and Conclusions

A series of experiments investigating spatiotemporal variability of the phytoplankton assemblage and potential effects of assimilating data from different observation platforms

highlighted the benefits and challenges of combining data and biogeochemical models in the Ross Sea. The assimilation of glider data reduced model-data misfit by 50%, and resulted in reduced depth-integrated primary production and higher carbon export at 200 m. Additional experiments for different spatial regions reduced predictive costs with respect to unassimilated data by ~35%, suggested that the model parameters were well constrained, and implied that

using glider data as time series in these local studies is a reasonable approach. This may further suggest the value of using moorings or buoys, or even deploying gliders in a "virtual mooring" mode. However, the effects of mesoscale variability were apparent when assimilating data at a frequency characteristic of cruise-based sampling, which resulted in a wide range of primary production and export estimates depending on the sampling times. Results of assimilating data

characteristic of satellite-based sampling suggest that assimilating satellite-derived data will result in underestimated carbon export. These findings can be used to help avoid potential sources of error when using ship-based or satellite-based observations alongside the development, calibration, or running of Ross Sea biogeochemical models. The combination of high-resolution glider data and modeling in this study underscores the importance of considering

how the timing at which observations are collected affect the subsequent interpretations, and it is believed that further comparison of data from additional autonomous gliders throughout the Ross Sea and between years will provide greater clarity of the spatiotemporal variability on multiple scales.



*Data Availability.* Data from the autonomous glider are available from the BCO-DMO data
repository (http://www.bco-dmo.org/dataset/568868), and other data to support this article
will be made available at W&M Publish (http://doi.org/xxxx) and upon request from the
authors (dekaufman@vims.edu, marjy@vims.edu).

*Competing Interests.* The authors declare that they have no conflict of interest.

*Acknowledgements.* This material is based upon work supported by the U.S. National Science
Foundation's Office of Polar Programs (NSF-ANT-0838980). The authors thank Drs.
Elizabeth A. Canuel, Eileen E. Hofmann, and Elizabeth H. Shadwick for constructive
comments. Additional thanks go to Michael S. Dinniman for helping with model forcings.
This work was performed (in part) using computational facilities at the College of William
and Mary which were provided by contributions from the National Science Foundation, the
Commonwealth of Virginia Equipment Trust Fund and the Office of Naval Research. This
paper is contribution XXXX of the Virginia Institute of Marine Science, College of
William and Mary.



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



**Table 1:** Eight parameters optimized in this analysis.

| Parameter Name | Initial value (Kaufman et al., 2017a) | Bounds (lower, upper) |
|---|---|---|
| **Diatom max growth rate at 0˚C** | 0.375 (d$^{-1}$) | 0.09375, 1.5 |
| *P. antarctica* **solitary cells C:Chl ratio** | 30 (gC gChl$^{-1}$) | 7.5, 120 |
| *P. antarctica* **colonies max growth rate at 0˚C** | 0.5 (d$^{-1}$) | 0.125, 2 |
| *P. antarctica* **solitary cells max growth rate at 0˚C** | 0.5 (d$^{-1}$) | 0.125, 2 |
| **Diatom C:Chl ratio** | 150 (gC gChl$^{-1}$) | 37.5, 600 |
| **Fast detritus sinking fraction of diatom losses** | 0.75 | 0.5, 0.95 |
| *P. antarctica* **colonies max sinking rate** | 20 (m d$^{-1}$) | 5, 80 |
| *P. antarctica* **colonies C:Chl ratio** | 40 (gC gChl$^{-1}$) | 10, 160 |



**Table 2:** Time, depth, and time-space resolution of glider-based observations of Chl and POC assimilated for each experiment.

| Experiment | Depth (m) | Temporal Resolution | Spatial Area(s) |
|---|---|---|---|
| **Expt 1a: Full Assimilation** | 0 - 50 | Hourly | Full glider track |
| **Expt 1b: Latitudinal Assim.** | 0 - 50 | Hourly | North, Central, South Latitudinal bands |
| **Expt 1c: Longitudinal Assim.** | 0 - 50 | Hourly | East, Central, West Longitudinal bands |
| **Expt 2a: Glider Assimilation** | 0 - 50 | ~ twice per day, separated at a minimum of 12 hours. | Full glider track |
| **Expt 2b: Cruise-based Assim.** | 0 - 50 | 3 days in a row, and then another 3 consecutive days two weeks later | Full glider track |
| **Expt 2c: Satellite-based Assim.** | 0 - 5 | 1 day every two weeks | Full glider track |



**Table 3:** Depth- and time-integrated primary production (PP), carbon export flux at 200 m, and costs for the No Assimilation run (cost = 0.77), Experiment #1 and #2.

| Simulation name | PP (g C m$^{-2}$ y$^{-1}$) | Export (g C m$^{-2}$ y$^{-1}$) | Predictive Cost ($\mathcal{J}_P$) | Assim. Cost ($\mathcal{J}_A$) |
|---|---|---|---|---|
| **No assimilation** | 111.7 | 18.8 | - | - |
| **Expt 1a: Full Assimilation** | 104.2 | 27.2 | - | 0.41 |
| **Expt 1b: Latitudinal Assim.** | 101.8[*] ±3.3 | 26.1 ±2.1 | 0.49 ±0.13 | 0.43 ±0.14 |
| **Expt 1c: Longitudinal Assim.** | 103.2 ±2.1 | 26.9 ±2.1 | 0.50 ±0.10 | 0.46 ±0.13 |
| **Expt 2a: Glider Assim.** | 103.7 ±1.8 | 27.0 ±1.2 | 0.43 ±0.01 | 0.43 ±0.03 |
| **Expt 2b: Cruise-based Assim.** | 113.1 ±22.3 | 24.8 ±6.6 | 1.24 ±0.95 | 0.52 ±0.19 |
| **Expt 2c: Satellite-based Assim.** | 114.1 ±10.7 | 16.7 ±2.7 | 1.04 ±0.36 | 0.26 ±0.16 |

[*] costs represent mean ± one standard deviation of assimilative runs.




**Table 4:** Initial parameter values (No Assimilation) and optimal parameter values after conducting the Full Assimilation, Glider, Cruise-based, and Satellite-based Assimilation experiments


| Parameter Name | Initial Value | Expt 1a Full Assimilation | Expt 2a Glider[*] | Expt 2b Cruise-based[*] | Expt 2c Satellite-based[*] |
|---|---|---|---|---|---|
| Diatom max growth rate at 0˚C (d$^{-1}$) | 0.375 | 0.40 | 0.43 ±0.01 | 0.42 ±0.15 | 0.41 ±0.09 |
| *P. antarctica* solitary cells C:Chl ratio (gC gChl$^{-1}$) | 30 | 29.7 | 25.84 ±5.16 | 37.3 ±26.7 | 51.5 ±26.8 |
| *P. antarctica* colonies max growth rate at 0˚C (d$^{-1}$) | 0.5 | 0.29 | 0.22 ±0.10 | 0.45 ±0.58 | 0.29 ±0.17 |
| *P. antarctica* solitary cells max growth rate at 0˚C (d$^{-1}$) | 0.5 | 0.39 | 0.45 ±0.06 | 0.75 ±0.70 | 0.79 ±0.51 |
| Diatom C:Chl ratio (gC gChl$^{-1}$) | 150 | 176.4 | 166.6 ±50.17 | 252.4 ±164.28 | 374.86 ±187.82 |
| Fast detritus sinking fraction of diatom losses | 0.75 | 0.87 | 0.86 ±0.05 | 0.86 ±0.11 | 0.62 ±0.14 |
| *P. antarctica* colonies max sinking rate (m d$^{-1}$) | 20 | 10.7 | 10.1 ±3.66 | 20.1 ±20.5 | 12.8 ±9.27 |
| *P antarctica* colonies C:Chl ratio (gC gChl$^{-1}$) | 40 | 14.0 | 14.2 ±2.29 | 42.7 ±41.6 | 34.3 ±26.5 |

[*] mean ± one standard deviation of assimilative runs.



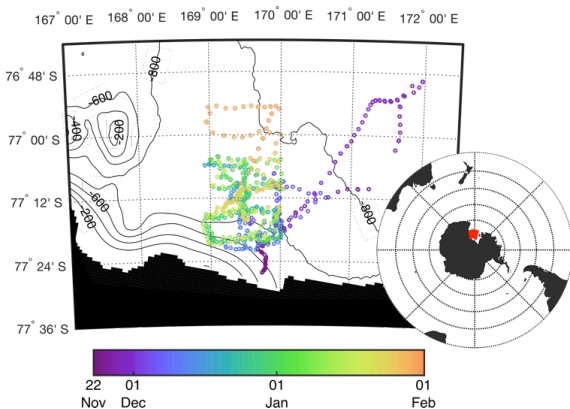

**Figure 1:** Southern Ross Sea showing transect locations where the glider was at the surface. The color of each glider dive indicates the date. Bathymetric contours are shown at 200-m intervals, as obtained from the bedmap2 bathymetric data [*Fretwell et al.*, 2013].





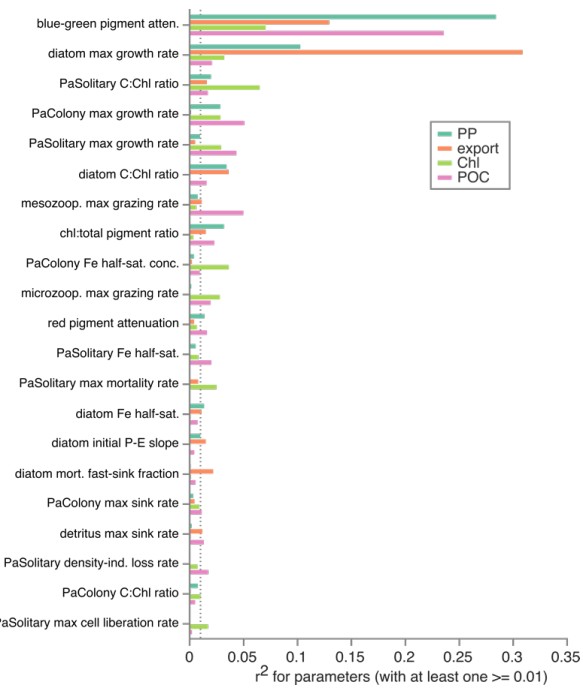

**Figure 2:** Variance explained in model outputs by parameters during sensitivity tests using Latin hypercube sampling of parameter space. Only parameters with at least one $r^2$ value greater than or equal to 0.01 (vertical dotted line) are shown.



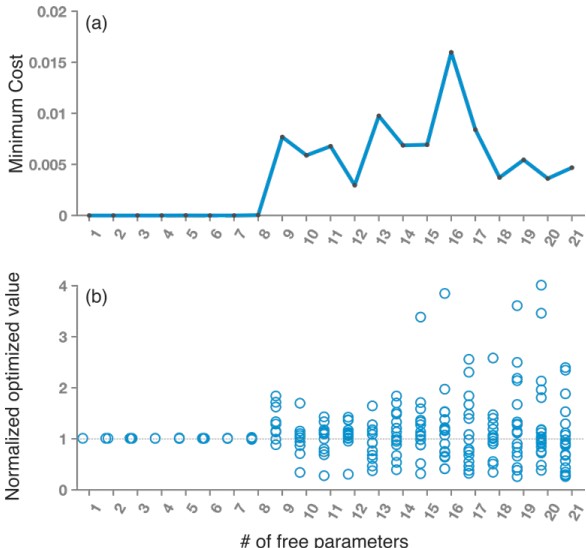

**Figure 3:** (a) Minimum costs and (b) normalized parameter values in numerical twin experiments, illustrating that the assimilation procedure is unable to successfully recover the true parameter values when more than eight parameters are optimized. One data point in three of the experiments (#s 19, 20, and 21) exceeds the y-axis upper limit in the lower (b) panel.





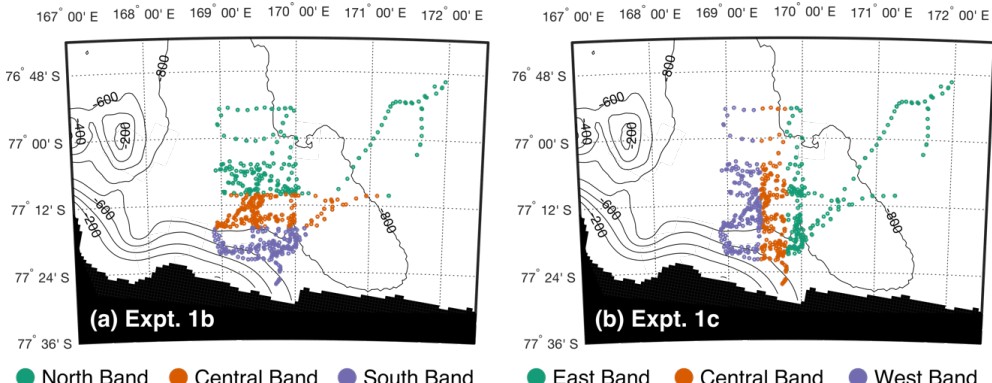

**Figure 4:** Locations of glider observations assimilated in (a) Experiment 1b –latitudinal bands, and (b) Experiment 1c – longitudinal bands. Colors represent the three spatial bands of data assimilated.



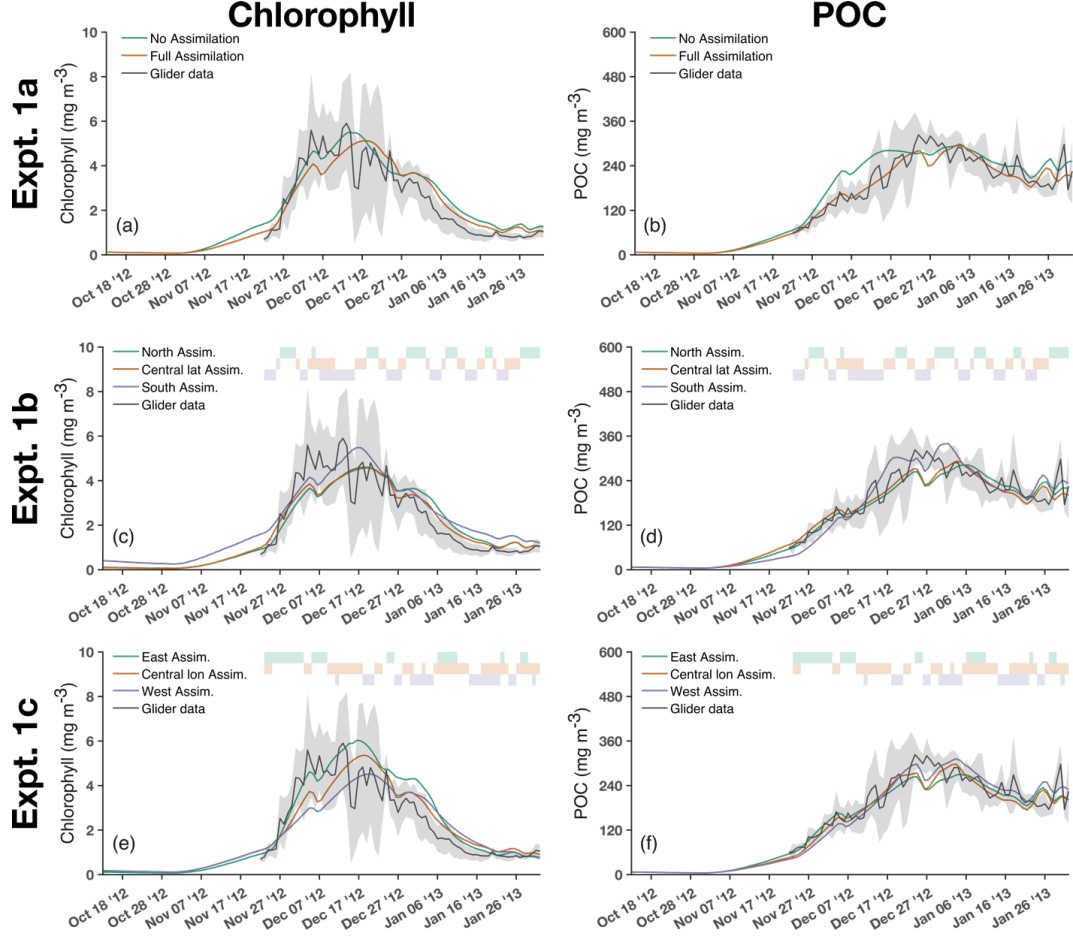

**Figure 5 (a-f):** Upper 50 m mean concentrations of (a,c,e) Chl and (b,d,f) POC for various experiments assimilating the full glider and from different spatial areas (Table 2): (a,b) Experiment 1a, (c,d) Experiment 1b - latitude bands, and (e,f) Experiment 1c - longitude bands. For reference, model results for the Full Assimilation case (orange lines), and glider data (black lines) with shading (gray) representing one standard deviation are included in each panel. Colored boxes at the top of each panel indicate times of assimilated observations.



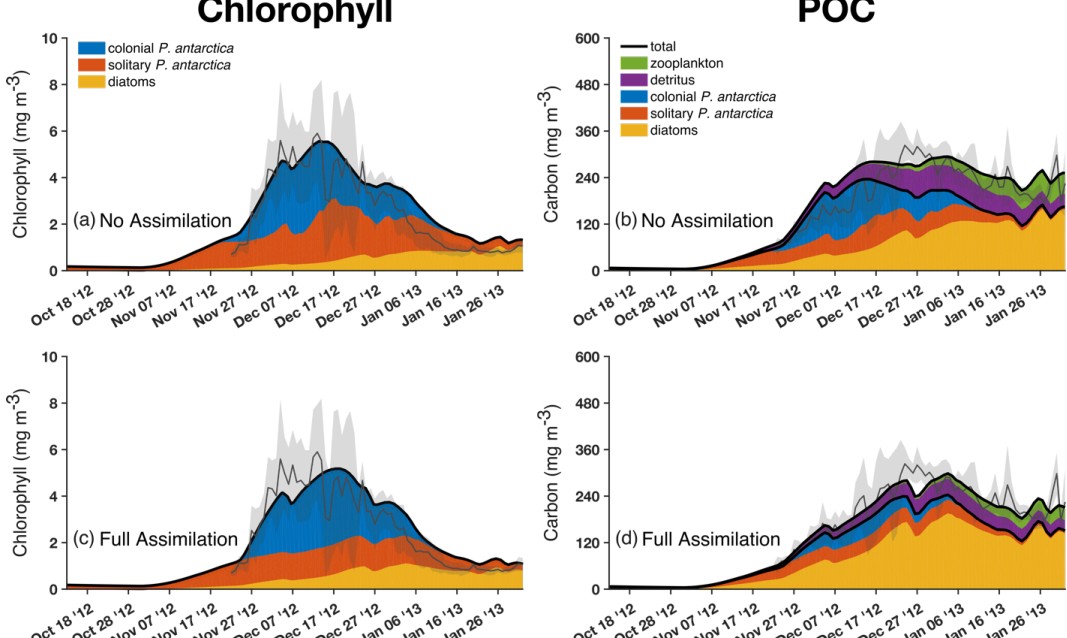

**Figure 6:** Upper 50 m mean concentrations of the three phytoplankton groups in terms of (a,c) Chl and (b,d) POC for the No Assimilation case (a,b) and the Full Assimilation case (c,d). The glider data are shown (black line) with shading (gray) that represents one standard deviation daily.





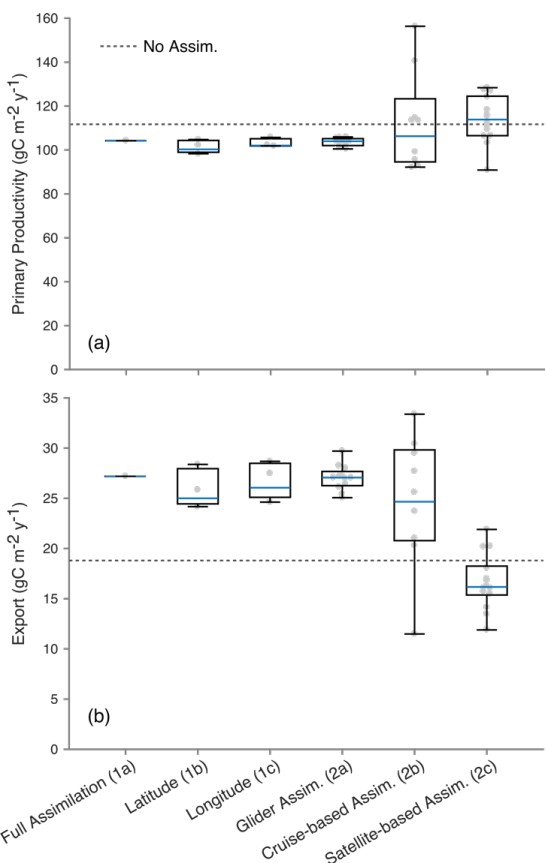

**Figure 7:** Distributions of (a) depth- and time-integrated production and (b) carbon export flux at 200 m for each assimilation experiment (Table 2). The median value for each experiment is indicated by a horizontal light-blue line. Each box extends vertically from the 1st to 3rd quartile, and the whiskers extend from the lowest to highest values. Individual values are shown as grey dots. For reference, production and export estimates from the No Assimilation (solid blue line) and Full Assimilation (dashed gray line) cases are included in each panel.





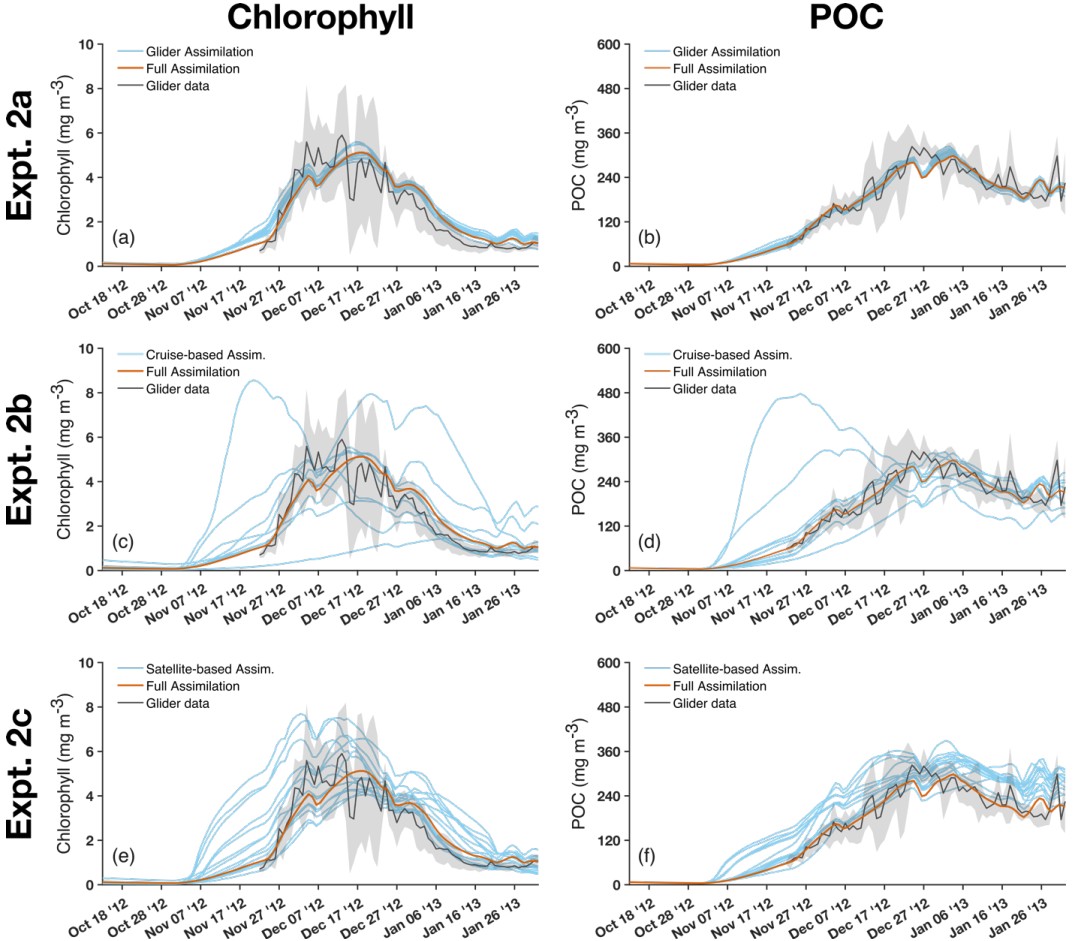

**Figure 8 (a-f):** Upper 50 m mean concentrations of (a,c,e) Chl and (b,d,f) POC for various experiments assimilating subsets characteristic of the original glider data, cruise-based observations and satellite-based observations (Table 2): (a,b) Experiment 2a - glider observations, (c,d) Experiment 2b - cruise-based, and (e,f) Experiment 2c - satellite-based. For reference, model results for the Full Assimilation case (orange lines), and glider data (black lines) with shading (gray) representing one standard deviation are included in each panel.