# Peer review of "Assimilating bio-optical glider data during a phytoplankton bloom in the southern Ross Sea"

_Biogeosciences, 2017_

## Referee Comment (RC1) · M.E. Gharamti (Referee) · 1 Sep 2017

The article presents a DA study in the Ross Sea, a region of the southern ocean. The authors use bio-optical glider data to reduce the model-data misfit of Chl concentration and POC. In the process, 8 different uncertain parameters are identified and optimized with incoming observations. The authors provide a thorough assessment of the DA system by changing the spatial and temporal resolution of the observations. This is performed in an effort to understand the impact of the number and type of observations (e.g., cruise and satellite) on the resulting biogeochemical modeling skill.

I think the paper is well-written, clear and nicely organized. The authors tackle an interesting problem that researchers within the DA-marine ecology community have

been investigating for a while. Although the results from such a small domain and a 1D model can not be generalized for large-scale problems (the authors recognize this), the article presents novel research points especially those of the parameter optimization. I have few minor comments (below), otherwise I don't see any reason for not publishing this article. It would be good to address the comments below in the manuscript before publishing.

1- Section 2.2: I would like to see how the observational error variance is parameterized. I believe the observational mapping operator is quite nonlinear. So, what procedure did the authors follow to find both $\sigma\_chl^2$ and $\sigma\_poc^2$ (twin and real experiments)?

2- Maybe I missed it but it would be good to provide a discussion on the computational cost of the genetic algorithm. Obviously, the authors are using some kind of hybrid algorithm (genetic + Powell) but I'm pretty sure these (non-gradient based) won't be as useful in large scale models. For instance, if the biogeochemical parameters are spatially varying then the degrees of freedom in the system will significantly increase. I'm not so much familiar with the algorithm the authors are using, so it would be good to see how does it compare computationally to an EnKF for example.

3- Section 2.4: I know it's mentioned somewhere, but it would be good to state that the algorithm selects random parameters within a range. After all, the chosen parameters need to be physically meaningful.

4- Lines 168-170: I am not sure what the authors mean by this sentence. Consider rephrasing.

5- Section 2.5.2: I think adding a small appendix section summarizing the differences between a MC and a Latin Hypercuble sampling would be useful for the reader.

6- Section 2.5.3: Why optimizing more parameters (>8) was not successful? Any reason for this, statistical one perhaps? Is it because the parameters maybe spatially

varying and this assumption is relaxed in the objective function? Or could it be due to the choice of the observational error variance? On another note, how to make sure it's not a drawback from the optimization algorithm itself? A paragraph addressing this is needed here. I could not find an explanation for such a behavior myself.

───────────────────────────

---

## Referee Comment (RC2) · Anonymous Referee #2 · 5 Sep 2017

The comment was uploaded in the form of a supplement:
https://www.biogeosciences-discuss.net/bg-2017-258/bg-2017-258-RC2-supplement.pdf

---

## Author Comment (AC1) · 4 Oct 2017

Dear M.E. Gharamti,

We greatly appreciate your valuable and constructive comments, which will help us improve and clarify the manuscript and presentation of the parameter selection and optimization procedures in particular. In response to your questions and suggestions, please find our answers and proposed changes (in blue) following each of your comments below. All line numbers refer to the original submitted manuscript.

We thank you again for your review.

Sincerely,
Daniel Kaufman, Marjorie Friedrichs, John Hemmings, Walker Smith

The article presents a DA study in the Ross Sea, a region of the southern ocean. The authors use bio-optical glider data to reduce the model-data misfit of Chl concentration and POC. In the process, 8 different uncertain parameters are identified and optimized with incoming observations. The authors provide a thorough assessment of the DA system by changing the spatial and temporal resolution of the observations. This is performed in an effort to understand the impact of the number and type of observations (e.g., cruise and satellite) on the resulting biogeochemical modeling skill.

I think the paper is well-written, clear and nicely organized. The authors tackle an interesting problem that researchers within the DA-marine ecology community have been investigating for a while. Although the results from such a small domain and a 1D model can not be generalized for large-scale problems (the authors recognize this), the article presents novel research points especially those of the parameter optimization. I have few minor comments (below), otherwise I don't see any reason for not publishing this article. It would be good to address the comments below in the manuscript before publishing.

Thank you for your positive comments.

1- Section 2.2: I would like to see how the observational error variance is parameterized. I believe the observational mapping operator is quite nonlinear. So, what procedure did the authors follow to find both \sigma_chl^2 and \sigma_poc^2 (twin and real experiments)?

The procedure we used was simply to calculate the inverse of the standard deviation, similar to other assimilation efforts (e.g. in Experiment #1 in Hemmings and Challenor, 2012; Friedrichs et al., 2006; Xiao and Friedrichs, 2014b). This definition of sigma was used for both the twin experiments described in section 2.5.3 (where the sigma was

calculated from the synthetic dataset) as well as the real assimilation experiments described in section 2.6 (where sigma was calculated from the particular observation set assimilated in each case).

You may notice that the second reviewer raised a similar question, and we there also indicate our proposed text for section 2.3, referencing other assimilation studies that have used standard deviations to weight the misfit contributions. Specifically, we propose modifying the text on lines 126-128 as follows: "where $N$ is the number of observation points, $x_i$ is the simulated value of either chlorophyll or POC at the $i$th observation point and $y_i$ is its observed value; $\sigma$ is the standard deviation of the specific observation set assimilated in a particular experiment. Using the standard deviation of the observations to define a characteristic scale of variation for each variable is a technique used in previous studies (e.g. Friedrichs et al., 2006; Xiao and Friedrichs, 2014). It is designed to weight the relative misfit contribution of each variable appropriately when there are insufficient data to define a comprehensive error model. Such a model would require reliable information about the uncertainty associated with observation errors (instrument error and error of representativeness) and non-parametric errors in the simulation such as forcing errors (Schartau et al., 2017). The use of different cost function weighting schemes in plankton modelling including the characteristic scale technique is explored in more detail by Hemmings and Challenor (2012)."

2- Maybe I missed it but it would be good to provide a discussion on the computational cost of the genetic algorithm. Obviously, the authors are using some kind of hybrid algorithm (genetic + Powell) but I'm pretty sure these (non-gradient based) won't be as useful in large scale models. For instance, if the biogeochemical parameters are spatially varying then the degrees of freedom in the system will significantly increase. I'm not so much familiar with the algorithm the authors are using, so it would be good to see how does it compare computationally to an EnKF for example.

To facilitate comparison with other assimilation methods, including EnKF, we propose including the number of model evaluations (approximately 4000 - 5000) in the text as described below. The number of evaluations and therefore the computational cost for our method is typically higher than EnKF, because our method is designed for a more comprehensive investigation of the parameter space. We think including computational cost in terms of run times in the manuscript would be of limited value because it would then also be necessary to report all hardware specs, which may be too much info and regardless, the hardware will probably be out-of-date in just a few years.

You are correct that these optimization methods won't be as useful in large-scale models when applied directly. However, the parameters identified in a 1D model by these techniques can be used in larger 3D models, and this has been shown to improve those larger models (e.g. see the review in section 7.2 of Schartau et al., 2017). Furthermore, you are correct to point out that allowing parameters to vary spatially would increase the degrees of freedom in the system with further implications for the practicality of our method. However, the method is not intended for estimating spatially varying parameters but for estimating parameters that are spatially uniform over as large a domain as

possible. Given the increased model degrees of freedom associated with spatially varying parameters and the consequent increased risk of over-fitting, it is unclear to what extent allowing parameters to vary spatially would be useful. The issue is discussed in detail in Schartau et al (2017).

We propose adding a paragraph at the end of section 4.1 to make the above points clear to the reader: "The high number of model evaluations in each optimization case (roughly 4000 – 5000) makes such direct optimization impractical for large-scale models; however, the parameters identified in a 1D model by these techniques can be used in larger models, and indeed locally optimized parameters have been previously shown to improve the skill of 3D models in other regions (Oschlies and Schartau, 2005; Kane et al., 2011; McDonald et al., 2012; St-Laurent et al., 2017). It is expected that the optimized parameter values found in the one-dimensional assimilation experiments described here will be of value in a future 3D biogeochemical modeling analysis of the Ross Sea and, through model inter-comparisons, provide a basis for examining the dependence of these parameter values on model structure and level of complexity, as has been done elsewhere (Friedrichs et al., 2007; Bagniewski et al., 2011; Ward et al., 2013; Irby et al., 2016)."

3- Section 2.4: I know it's mentioned somewhere, but it would be good to state that the algorithm selects random parameters within a range. After all, the chosen parameters need to be physically meaningful.

This is a good point, and yes, the selection of values from within the range for each parameter is discussed in section 2.5.1. Nevertheless, it is understandable for the reader to be wondering about it earlier in section 2.4.  To help the reader, we propose adding a sentence at line 147 that provides a brief clarification and cross-references section 2.5.1: "The constituent parameter values are selected randomly from within a pre-determined range of allowable values (Sect. 2.5.1)."

4- Lines 168-170: I am not sure what the authors mean by this sentence. Consider rephrasing.

In order to clarify the meaning of this sentence, we propose rephrasing it to: "Ideally, optimal values are identified for all parameters in a model, however, uncertainty in the parameter estimates from an algorithmic optimization increases as the number of parameters included in that optimization increases (Friedrichs et al., 2007; Ward et al., 2010)."

5- Section 2.5.2: I think adding a small appendix section summarizing the differences between a MC and a Latin Hypercube sampling would be useful for the reader.

This is a good idea. We propose adding an appendix titled "Appendix A: Latin hypercube sampling (Sect. 2.5.2)" and inserting a cross-reference to this appendix on line 199. The proposed appendix would have the following suggested text:

"Latin hypercube sampling (LHS) and Monte Carlo sampling are both techniques that can be used to randomly draw a finite number of samples from input distributions in order to approximate a full multidimensional distribution. The LHS incorporates stratified random sampling, i.e. in each dimension each sample is drawn randomly from within a different interval (also called a stratification or layer) of the distribution (McKay et al., 1979). Intervals are chosen with reference to the probability distribution such that each represents an equally probable range. In contrast, Monte Carlo sampling proceeds in each dimension with each sample drawn randomly from the entire distribution. Stratified random sampling with intervals of uniform probability ensures a good representation of the distribution, reducing the risk of samples being clustered in one or a small number of areas. In LHS sampling, if the sample size is n, each dimension is divided into n intervals such that in multi-dimensional space each interval of each dimension is sampled once and once only. This is based on the idea of a Latin square in which an individual symbol appears once in each row and each column. It ensures a good representation of the distribution is achieved for all dimensions."

6- Section 2.5.3: Why optimizing more parameters (>8) was not successful? Any reason for this, statistical one perhaps? Is it because the parameters maybe spatially varying and this assumption is relaxed in the objective function? Or could it be due to the choice of the observational error variance? On another note, how to make sure it's not a drawback from the optimization algorithm itself? A paragraph addressing this is needed here. I could not find an explanation for such a behavior myself.

The primary reason for unsuccessfully optimizing more parameters is that many model parameters are correlated. Previous studies have also followed a procedure to reduce the set of optimizable parameters, particularly to avoid optimizing highly correlated parameters. For examples of these procedures, see Xiao and Friedrichs, 2014b, Friedrichs et al. 2006 (or 2007).

To help any reader wondering similarly about limiting the number of optimized parameters and to clarify that this is not a drawback of this specific optimization algorithm, we propose adding the following paragraph in section 2.5.3 (line 222): "There is a limit to the number of parameters that can be independently constrained by the available observations because varying different parameters can often have similar effects on the cost function. Optimizing a larger set increases the potential for correlation between the effects of different parameters, reducing the algorithm's effectiveness in identifying unique optimal parameter sets. This, combined with the increased potential for over-fitting associated with the greater model degrees of freedom, can reduce the ability of an optimized model to reproduce independent data sets (Matear et al., 1995; Friedrichs et al., 2007; Xiao and Friedrichs et al., 2014b). The limitation on the number of optimizable parameters applies to both μGA and variational adjoint optimizations (Ward et al., 2010). In fact, rather than being a function of the optimization algorithm, it is dependent on the available data and the design of the cost function. A larger or richer observation set can help to constrain more parameters. The impact of cost function design is more complicated because an improved cost function may allow for greater uncertainty

in the observations and/or non-parametric uncertainty in the simulation, leading to weaker but more realistic constraints on the parameters (Hemmings & Challenor, 2012)."

In addition, we propose modifying the first sentence of section 2.5.3 (line 213) to read: "After selecting the 21 potentially optimizable parameters, Numerical Twin Experiments (NTEs) were conducted to identify an optimizable subset by evaluating…"

Additional literature cited in responses:

Bagniewski, W., Fennel, K., Perry, M. J., and D'Asaro, E. A.: Optimizing models of the North Atlantic spring bloom using physical, chemical and bio-optical observations from a Lagrangian float, *Biogeosciences*, *8*(5), 1291–1307, doi:10.5194/bg-8-1291-2011, 2011.

Irby, I. D., Friedrichs, M. A. M., Friedrichs, C. T., Bever, A. J., Hood, R. R., Lanerolle, L. W. J., Li, M., Linker, L., Scully, M. E., Sellner, K., Shen, J., Testa, J., Wang, H., Wang, P., Xia, M.: Challenges associated with modeling low-oxygen waters in Chesapeake Bay: a multiple model comparison, *Biogeosciences*, *13*(7), 2011–2028, doi:10.5194/bg-13-2011-2016, 2016.

Kane, A., Moulin, C., Thiria, S., Bopp, L., Berrada, M., Tagliabue, A., Crépon, M., Aumont, O., and Badran, F.: Improving the parameters of a global ocean biogeochemical model via variational assimilation of in situ data at five time series stations, *J. Geophys. Res. Ocean.*, *116*(6), 1–14, doi:10.1029/2009JC006005, 2011.

McDonald, C. P., Bennington, V., Urban, N. R., and McKinley, G. A.: 1-D test-bed calibration of a 3-D Lake Superior biogeochemical model, *Ecol. Modell.*, *225*, 115–126, doi:10.1016/j.ecolmodel.2011.11.021, 2012.

McKay, M. D., Beckman, R. J., and Conover, W. J.: A Comparison of Three Methods for Selecting Value of Input Variables in the Analysis of Output from a Computer Code, *Technometrics*, *21*(2), 239–245, 1979.

Oschlies, A., and Schartau, M.: Basin-scale performance of a locally optimized marine ecosystem model, *J. Mar. Res.*, *63*(2), 335–358, doi:10.1357/0022240053693680, 2005.

St-Laurent, P., Friedrichs, M.A.M., Najjar, R.G., Martins, D.K., Herrmann, M., Miller, S.K., and Wilkin, J.: Impacts of atmospheric nitrogen deposition on surface waters of the western North Atlantic mitigated by multiple feedbacks. *J. Geophys. Res. Ocean.*, in press September 2017.

Thomalla, S. J., Racault, M., Swart, S., and Monteiro, P. M. S.: High-resolution view of the spring bloom initiation and net community production in the Subantarctic Southern Ocean using glider data, *ICES J. Mar. Sci. J. du Cons.*, *72*(6), 1999–2020, doi:10.1093/icesjms/fsv105, 2015.

---

## Author Comment (AC2) · 4 Oct 2017

Dear Reviewer #2,

We greatly appreciate your time and effort spent reviewing our manuscript. According to your constructive feedback, we propose changes to clarify aspects of model setup, optimization method, and conclusions. Please find our responses (in blue) following each of your comments below. All line numbers refer to the original submitted manuscript. Thank you again for your review.

Sincerely,
Daniel Kaufman, Marjorie Friedrichs, John Hemmings, Walker Smith

Review comments for the manuscript: "Assimilating bio-optical glider data during a phytoplankton bloom in the southern Ross Sea (bg-2017-258)" by Daniel E. Kaufman, Marjorie A. M. Friedrichs, John C. P. Hemmings, and Walker O. Smith Jr.

The authors present a data assimilation study that optimizes parameters in a one-dimensional biogeochemical model using glider observations in the southern Ross Sea. They show insensitivity of the result to the geographical location of observations, but the optimizing parameters is sensitive to the sampling frequency.

The paper is overall well-written, but I hope the reviewers be able to address comments that I have.

– The procedure can be clarified more. This study utilizes a one-dimensional model for 3D observations. Does the cost function use all the observations and estimate one set of the parameter? Or is there an optimized parameter set for each location? If the first approach is used, do you expect that the optimized parameter values represent the distribution of those obtained by the second approach?

As described on line 258 and in Table 2, the cost function uses all of the observations for experiment #1A. To make this clearer, we propose modifying the sentence in the abstract on line 16 to read "Assimilation of data from the entire glider track …"

We do find that the optimized parameter values from the first approach (using all observations) represent those obtained by the second approach (using observations from different locations), as described in the latitudinal (Expt. 1b) and longitudinal (Expt. 1c) experiments and shown in Table 4.

– The authors argue that the data assimilation performance is sensitive to the observation sampling frequency due to "mesoscale variability". Mesoscale variability also means the

variation in space with the scale of O(100km). But it is odd to see that the geographical region does not show a big impact on the performance. Could the author comment on this?

Here we define mesoscale variability as "days-weeks, 1-10 km" (line 59). We had no a priori expectation that the geographical regions would show minor differences in model solutions, however we believe that the minor differences are reasonably explained in section 4.2, and especially in this section's last paragraph. Although in situ observations from previous studies have shown spatial differences on these scales, it has been unclear whether the differences were due to temporal or spatial variations. The assimilation experiments in this study suggest that variability observed on the mesoscale in this geographical region may be more likely due to temporal patterns than spatial differences. Therefore, one could expect that assimilating these different locations would show a bigger impact if the observation times concurrently varied, such as is demonstrated in the cruise-based and satellite-based assimilation cases. On larger scales, however, it is likely that the importance of spatial variability would be greater. For instance, the distinct spatial differences observed by satellites are generally across scales larger than the 1-10 km discussed here.

– By construction, the role of advection is not considered in this study. Can authors comment on the role of advection in this region? Do authors think the insensitivity of the assimilation performance to the geographical location of observations is related to the omission of advection?

Previous studies have suggested that horizontal transport and eddies may be important near island land masses and the Ross Ice Shelf (Gerringa et al., 2015; Li et al., 2017). In this region of the Ross Sea particularly, moorings and modeling have indicated moderate westward currents close to the ice shelf (Keys et al., 1990); nevertheless, advection appears to be weaker as one moves farther from the shelf edge (Dinniman et al., 2003).

One cannot rule out the possibility that the sensitivity of the optimizations to the observations' location could be affected by adding advection to the model. However, this would likely only be the case if there were, in reality, strong horizontal velocity gradients, i.e. differences in advection between the observation locations. A more thorough examination of the role of horizontal advection on modeled dynamics of the phytoplankton assemblage is beyond the scope of the current study, but would be benefitted greatly by contemporaneous and co-located mooring and/or ship-based current measurements.

– line 85. The effort on estimating biological state variables can be listed here. (e.g., Song, H., C. A. Edwards, A. M. Moore and J. Fietcher, 2016: Data assimilation in a coupled physical-biogeochemical model of the California Current System using an incremental lognormal 4-dimensional variational approach: Part 3, Assimilation in a realistic context using satellite and in situ observations. Ocean Model., 106, 159-172.)

It is a good idea to reference Song et al. 2016 here.

– section 2.1: What is the vertical resolution of the model?

To clarify this, we propose adding details of the model setup to section 2.1, on line 108: "The model is configured to focus on dynamics within the euphotic zone with a vertical resolution of 5 m from the ocean surface to 200 m."

– line 114: The full name of BCO-DMO can be given.

Absolutely. We will expand the acronym to the full name.

– line 115: 5-m vertical binning is done using averages? or weighted average?

Vertical binning of the glider data was accomplished using averages, and to clarify this we propose modifying the sentence to read: "Data spanning the upper 200 m of the water column were binned by means into hourly, 5-m vertical bins.

– Equation for the cost function shows that the observational error covariance is estimated using the standard deviation of the observations. Is this right? I think using standard deviation may overestimate the observational error if the blooms dominate the chloro- phyll variability. If the error levels of the instruments are known, why not use these values?

The misfit contributions are weighted by using the inverse of the standard deviation, similar to other assimilation efforts (e.g. in Experiment #1 in Hemmings and Challenor, 2012; Friedrichs et al., 2006; Xiao and Friedrichs, 2014). If the aim were to estimate observational error, then the increase in variance due to the bloom would indeed likely lead to over-estimation. However the aim here is to weight the misfit contributions of chlorophyll and POC, and there is less impact of the bloom on these relative weights. Generally, a more sophisticated treatment of uncertainty in both the observations and the model is desirable as indicated by Hemmings & Challenor (2012), but such a treatment is beyond the scope of the present study and may not be practical with the available data. It makes sense therefore to initially employ a simple well-established method as we have done, but we recognize that it does have its limitations.

You may notice that the first reviewer raised a similar question as well, and we there also indicate our proposed text for section 2.3, referencing other assimilation studies that have used standard deviations to weight the misfit contributions. Specifically, we propose modifying the text on lines 126-128 as follows: "where $N$ is the number of observation points, $x_i$ is the simulated value of either chlorophyll or POC at the $i$th observation point and $y_i$ is its observed value; $\sigma$ is the standard deviation of the specific observation set assimilated in a particular experiment. Using the standard deviation of the observations to define a characteristic scale of variation for each variable is a technique used in previous studies (e.g. Friedrichs et al., 2006; Xiao and Friedrichs, 2014). It is designed to weight the relative misfit contribution of each variable appropriately when there are insufficient data to define a comprehensive error model. Such a model would require reliable

information about the uncertainty associated with observation errors (instrument error and error of representativeness) and non-parametric errors in the simulation such as forcing errors (Schartau et al., 2017). The use of different cost function weighting schemes in plankton modelling including the characteristic scale technique is explored in more detail by Hemmings and Challenor (2012)."

– section 2.4: Personally, it is not easy to digest this method. Maybe a diagram can help me and readers to understand the assimilation procedure better.

We appreciate the difficulty in understanding this section without a high level overview. Although we do not believe a full diagram is necessary, we propose two changes to this section to offer the reader a broader view of the method, rather than its current focus on technical details.

First, to clarify what is being done, rather than how, we propose changing the title of this section from "Implementation of micro-genetic algorithm and direction set algorithm" to "Cost function minimization."

Second, we propose adding a paragraph to the beginning of this section that summarizes the role of the two algorithms:
"Model parameters were optimized in MarMOT by finding the minimum of the cost function (Sect. 2.3) through a combination of the micro-genetic algorithm (μGA) and Powell's non-gradient direction set algorithm. The μGA runs first and identifies sets of parameter values that produce low cost values; this is achieved by "evolving" a population of various parameter sets over successive iterations, called generations. The low-cost parameter sets identified by the μGA are then used as starting points for the direction set method, which performs successive linear searches to identify nearby lower cost solutions."

– lines 244–245: Can you provide the number for the difference? If these two cases (50 m vs 200 m) are not significantly different, I would rather present the one with 200 m. Is it because of the computational time? (Also I hope the authors say something about the speed of this data assimilation calculation).

There is a relatively minor (~14%) difference between the results of the assimilation down to 50 m compared to 200 m. The trends and major conclusions of the study are likely not strongly affected by this choice. Conducting the assimilations for the upper 50 m avoided issues related to assimilating many low values of chlorophyll and POC, and also enabled a direct comparison of these results with the results of Kaufman et al. (2017) who similarly focused on the upper 50m concentrations. Computational time did not play a role in the decision to present results for the upper 50 m.

In further response to your question about computational cost, along with reviewer #1, we propose adding the number of model evaluations conducted for the assimilation experiments to the end of section 4.2, with the text: "The high number of model

evaluations in each optimization case (roughly 4000 – 5000) makes such direct optimization impractical for large-scale models; however, the parameters identified in a 1D model by these techniques can be used in larger models, and indeed locally optimized parameters have been previously shown to improve the skill of 3D models in other regions [Oschlies and Schartau, 2005; Kane et al., 2011; McDonald et al., 2012; St-Laurent et al., 2017]."

– section 2.6.2: Are there any changes in spatial coverage between "glider", "cruise" and "satellite" data cases? If they have the same spatial coverages, naming this way may confuse readers because it is obvious that their spatial coverages are significantly different.

As mentioned (on line 418), these cases alter both spatial and temporal resolution, and therefore they don't have identical spatial coverage. As such, we feel these names are appropriate.

– lines 474–476: Do authors have any ideas why satellite-derived data underestimates carbon export?

This is addressed earlier in the manuscript on line 424: "The lower estimates of carbon export occurred because the optimal diatom fraction for fast-sinking detritus obtained via the assimilation of surface-only data ($0.62 \pm 0.14$) was significantly lower than that obtained via the assimilation of data throughout the upper 50 m (Expt. 2a: $0.86 \pm 0.05$; Expt. 2b: $0.86 \pm 0.11$)."

– lines 480–483: I think the phrases after "and it is" are not necessary. Please consider to remove them.

Excellent idea. We agree and will take out the phrase starting with "and it is", and we will also remove the unnecessary "Ross Sea" on line 478.

Additional literature cited in responses:

Dinniman, M. S., Klinck, J. M., and Smith, W. O.: Cross-shelf exchange in a model of the Ross Sea circulation and biogeochemistry, *Deep-Sea Res. II*, 50(22–26), 3103–3120, doi:10.1016/j.dsr2.2003.07.011, 2003.

Gerringa, L. J. A., Laan, P., van Dijken, G. L., van Haren, H., De Baar, H. J. W., Arrigo, K. R., and Alderkamp, A.-C.: Sources of iron in the Ross Sea Polynya in early summer, *Mar. Chem.*, *177*, 447–459, doi:10.1016/j.marchem.2015.06.002, 2015.

Kane, A., Moulin, C., Thiria, S., Bopp, L., Berrada, M., Tagliabue, A., Crépon, M., Aumont, O., and Badran, F.: Improving the parameters of a global ocean

biogeochemical model via variational assimilation of in situ data at five time series stations, *J. Geophys. Res. Ocean.*, *116*(6), 1–14, doi:10.1029/2009JC006005, 2011.

Keys, H. (J. R.), Jacobs, S. S., and Barnett, D.: The calving and drift of iceberg B-9 in the Ross Sea, Antarctica, *Antarct. Sci.*, *2*(3), 243–257, doi:10.1017/S0954102090000335, 1990.

McDonald, C. P., Bennington, V., Urban, N. R., and McKinley, G. A.: 1-D test-bed calibration of a 3-D Lake Superior biogeochemical model, *Ecol. Modell.*, *225*, 115–126, doi:10.1016/j.ecolmodel.2011.11.021, 2012.

Oschlies, A., and Schartau, M.: Basin-scale performance of a locally optimized marine ecosystem model, *J. Mar. Res.*, *63*(2), 335–358, doi:10.1357/0022240053693680, 2005.

St-Laurent, P., Friedrichs, M.A.M., Najjar, R.G., Martins, D.K., Herrmann, M., Miller, S.K., and Wilkin, J.: Impacts of atmospheric nitrogen deposition on surface waters of the western North Atlantic mitigated by multiple feedbacks. *J. Geophys. Res. Ocean.*, in press September 2017.

---

## Author Response (AR2)

Comments to the Author:

Dear Dr. Kaufman,

Two reviewers have reviewed the revised version of your manuscript, "Assimilating bio-optical glider data during a phytoplankton bloom in the southern Ross Sea", and your response to their comments. They both found that you adequately addressed their concerns and I agree with them. Based on that I suggest that your manuscript be accepted for publications in Biogeosciences provided that you address the minor comments here below.

Thank you for submitting your manuscript to Biogeosciences.

With kind regards

Marilaure Grégoire

Dear Marilaure Grégoire,

Thank you very much for your careful review of our manuscript, and for providing detailed comments. Please find below our manuscript changes and responses (in blue) to your comments.

Sincerely,

Daniel Kaufman, Marjorie Friedrichs, John Hemmings, Walker Smith

Comments:

Line 19: please correct the units and add a minus in front of the exponent "2". It should be 104 g C m-2 y-1

Corrected.

Line 65; please mention after Phaeocystis Antarctica, " (P.antarctica)"

Added.

Line 396: I would say "Phytoplankton growth and sinking rates" instead of "phytoplankton rates" because this does not concern all rates.

This line has been modified as suggested.

**Line 399: 'the initial hand-tuned simulation" You mean the "No Assimilation" experiment? "Optimal simulation" you mean "Full Assimilation" experiment ? Please clarify and uniformize throughout the manuscript in order to avoid confusion.**

Thank you for proposing this clarification. The text has been changed to say "Full Assimilation." and "No Assimilation." We have similarly clarified the text on lines 324 and 407, which now reference the "No Assimilation case" and "Full Assimilation," respectively.

Line 410: growth rates are not in m/day but /day. Please correct.

Corrected.

    Table 2, legend. Please clarify where the time is given. According to my understanding the first column refers to the type of experiments performed or simulation name as used in Table 3 and not to the Time as currently mentioned.

The wording has been simplified so that it neither duplicates the information in the column headers nor implies a one-to-one relationship with the column headers. Now it says "Spatio-temporal resolution of glider-based observations..."

Table 3, legend: Please clarify the period over which the time averaging has been performed (length of simulation and then converted into annual mean values?). I do not understand why the footnote ("costs represent means +/- one std") related to the cost is put for the first column. I would have put that in the legend for clarity. Finally, for helping the reader, I would briefly explain what are the
"predictive costs" and "assimilative costs" (a few words are enough)

    As suggested, we have added the parenthetical clarification, "(over the length of the simulation, representing yearly rates)". The footnote has been moved into the table caption with the following text to distinguish between the two costs: "Costs
provide a measure of the misfit between a particular model simulation and observations, and the costs shown represent mean ± one standard deviation of assimilative runs. The assimilative and predictive costs are computed from the assimilated and unassimilated data, respectively."

Figure 5: You mention "For reference, model results for the Full Assimilation case (orange lines), and glider data (black lines) with shading (gray) representing one standard deviation are included in each panel." However, the orange curve in each graph is not always the full assimilation case but rather the central lat/lon band assimilation case. Besides, please clarify how the standard deviation is computed.

To correct this, we have removed the unnecessary text so that the caption for Figure 5 (a-f) says: "
[revised manuscript text omitted]